# Characterization of Systemic and Culprit-Coronary Artery miR-483-5p Expression in Chronic CAD and Acute Myocardial Infarction Male Patients

**DOI:** 10.3390/ijms24108551

**Published:** 2023-05-10

**Authors:** Olga Volodko, Natalia Volinsky, Merav Yarkoni, Nufar Margalit, Fabio Kusniec, Doron Sudarsky, Gabby Elbaz-Greener, Shemy Carasso, Offer Amir

**Affiliations:** 1The Lydia and Carol Kittner, Lea and Benjamin Davidai Division of Cardiovascular Medicine and Surgery and Research Institute, Tzafon Medical Center, Affiliated with Azrieli Faculty of Medicine, Bar Ilan University, Tiberias 1528001, Israel; 2The Azrieli Faculty of Medicine, Bar-Ilan University, Safed 1311502, Israel; 3Heart Institute, Hadassah University Medical Center, Jerusalem, Department of Cardiology, Hadassah Medical Center, Faculty of Medicine, Hebrew University Jerusalem, Jerusalem 9574409, Israel

**Keywords:** miR-483-5p, next generation sequencing, myocardial infarction, coronary artery disease, culprit artery

## Abstract

Coronary artery disease (CAD) is the leading cause of mortality worldwide. In chronic and myocardial infarction (MI) states, aberrant levels of circulating microRNAs compromise gene expression and pathophysiology. We aimed to compare microRNA expression in chronic-CAD and acute-MI male patients in peripheral blood vasculature versus coronary arteries proximal to a culprit area. Blood from chronic-CAD, acute-MI with/out ST segment elevation (STEMI/NSTEMI, respectively), and control patients lacking previous CAD or having patent coronary arteries was collected during coronary catheterization from peripheral arteries and from proximal culprit coronary arteries aimed for the interventions. Random coronary arterial blood was collected from controls; RNA extraction, miRNA library preparation and Next Generation Sequencing followed. High concentrations of microRNA-483-5p (miR-483-5p) were noted as ‘coronary arterial gradient’ in culprit acute-MI versus chronic-CAD (*p* = 0.035) which were similar to controls versus chronic-CAD (*p* < 0.001). Meanwhile, peripheral miR-483-5p was downregulated in acute-MI and chronic-CAD, compared with controls (1.1 ± 2.2 vs. 2.6 ± 3.3, respectively, *p* < 0.005). A receiver operating characteristic curve analysis for miR483-5p association with chronic CAD demonstrated an area under the curve of 0.722 (*p* < 0.001) with 79% sensitivity and 70% specificity. Using in silico gene analysis, we detected miR-483-5p cardiac gene targets, responsible for inflammation (*PLA2G5*), oxidative stress (*NUDT8*, *GRK2*), apoptosis (*DNAAF10*), fibrosis (*IQSEC2*, *ZMYM6*, *MYOM2*), angiogenesis (*HGSNAT*, *TIMP2*) and wound healing (*ADAMTS2*). High miR-483-5p ‘coronary arterial gradient’ in acute-MI, unnoticed in chronic-CAD, suggests important local mechanisms for miR483-5p in CAD in response to local myocardial ischemia. MiR-483-5p may have an important role as a gene modulator for pathologic and tissue repair states, is a suggestive biomarker, and is a potential therapeutic target for acute and chronic cardiovascular disease.

## 1. Introduction

Alteration in gene expression patterns have been related to pathological cardiac conditions, which can lead to heart failure as we and others have previously described [1,2]. MicroRNAs (miRNAs) are key players in gene expression and protein regulation, and the scarcity or abundance of miRNAs in the cells may eventually lead to dysfunction of the tissues and ultimately to disease. Circulating miRNAs often reflect changes in miRNA expression in solid tissues and blood and their analysis can promote our understanding of different pathophysiological processes, including cardiovascular disease [3]. Different atherosclerotic manifestations, caused by genetic and environmental factors, contribute to chronic coronary artery disease (CAD) and eventually lead to acute myocardial infarction (MI), the leading cause of morbidity and mortality in developed countries [4,5,6].

The culprit vessel in an acute coronary occlusion encompasses a collection of cell types, which includes endothelial cells that line the inner wall and promote leukocyte adhesion, and smooth muscle cells that form the surrounding outer layer, modified by the atherosclerotic plaque. Pericytes wrap around endothelial cells and modulate the vascular permeability, tortuosity and coronary blood flow [7]. Several studies suggest that the formation of a culprit microenvironment displays very specific characteristics. Blood collected from a culprit area is enriched in activated, toll-like receptors 2 and 4 expressing monocytes and macrophages [8]. These cells were suggested to play a role in the inflammatory response and apoptosis, clearance of dead cells and the stimulation of the reparative processes, angiogenesis, and tissue repair, as well as cell regeneration [9]. In addition, elevated plasma levels of MMP-9 and IL-6 were detected at the site of the coronary plaque, and other inflammatory markers were systemically elevated in MI patients, such as IL-10, CRP, TNF-α and s-PLA2, while no specific elevation was detected in the culprit coronary artery [10].

The aim of the current study was to identify miRNAs associated with chronic CAD and acute MI manifestations (i.e., either with or without ST segment elevation, STEMI and NSTEMI, respectively) and to compare expression of miRNAs detected in peripheral blood vessels versus coronary arteries proximal to the culprit area.

## 2. Results

### 2.1. Patient Groups and Prevalence of Previous History

Patients included in this study were all men, aging 62 ± 9 years (range 40–88 years). All study groups included a mixed Jewish and Arab population without significant differences between the groups (*p* = 0.413 by ANOVA). The patients’ age and histories of hypertension, diabetes mellitus, hyperlipidemia, renal failure or previous atrial fibrillation were similar among the four groups: No CAD (control), CAD, NSTEMI, and STEMI (Table 1). Smoking was more prevalent in patients presenting with STEMI (16 of 22 patients, 72%, *p* = 0.002) while previous MI was more common in patients with NSTEMI (6 of 26 patients, 24%, *p* = 0.01). Blood count and renal function were in the normal range and did not differ among the independent groups.

Aortic stenosis patients were not excluded in this study. We had three patients with patent arteries and aortic stenosis (13%), and another three patients with chronic CAD and aortic stenosis (11%). There was no correlation between miR-483-5p expression and aortic stenosis in peripheral nor in coronary blood. The three individuals who showed aortic stenosis in the patent group did not affect the differential expression of miR-483-5p between the groups.

### 2.2. MiRNA Expression in Chronic CAD and Acute MI

To identify miRNA association with chronic CAD and acute MI, next generation sequencing (NGS) was performed on serum samples from 16 patent and 18 STEMI patients. Fifteen miRNAs identified as significantly up- or downregulated in STEMI compared with patent patients, as shown in Appendix A. In addition to the miRNAs already reported to be strongly associated with cardiovascular conditions, such as miR-1-3p and miR-150-5p [11,12,13,14,15,16], we also identified several miRNAs that are less recognized in CAD. We selected miR-483-5p for further validation by qPCR, since it demonstrated the next most significant difference (0.0041) after miR-1-3p and miR-150-5p with relatively high expression, suggesting its potential high impact in CAD pathogenesis (Appendix A).

### 2.3. MiR-483-5p Regulation in Chronic CAD and Acute MI

High concentrations of miR-483-5p were observed as ‘coronary arterial gradient’ in the culprit artery, the infract-related artery in acute-MI, compared with chronic-CAD artery. Specifically, we detected a significantly higher ‘coronary arterial gradient’ of miR-483-5p in STEMI patients compared to CAD patients (*p* = 0.035) locally at the culprit vessel. Moreover, the ‘coronary arterial gradient’ of miR-483-5p was significantly higher in NSTEMI/STEMI manifestations similar to controls when it was compared to chronic CAD patients (*p* < 0.001). When comparing the four groups by one-way analysis of variance (ANOVA, Kruskal–Wallis test), miR-483-5p in the culprit vessel was significantly lower in patients with chronic CAD (as demonstrated by stable angina pectoris) compared to all other clinical patterns (*p* = 0.0008; Figure 1B). At the periphery, miR-483-5p expression in circulating blood of chronic CAD and acute STEMI/NSTEMI patients was significantly lower compared with miR-483-5p expression in the patent group 1.1 ± 2.2 vs. 2.6 ± 3.3, respectively, *p* = 0.0013; Figure 1A) with no pathogenesis. In a stepwise logistic regression, only two parameters remained independently associated with CAD clinical patterns. Smoking was associated with a 4.4-fold increased risk of chronic CAD and acute MI (*p* < 0.02), and a 17% risk of CAD clinical patterns requiring percutaneous coronary intervention (PCI) was associated with peripheral high expression of miR-483-5p (as demonstrated by 0.83 odds ratio, *p* < 0.04, Table 2). Repeating this analysis in patients without prior history chronic CAD, previous acute MI or PCI produced similar results (smoking 6.7-fold increase, *p* < 0.005; 12% risk, *p* < 0.05).

ROC analysis for peripheral miR-483-5p association with chronic CAD demonstrated an area under the curve of 0.713 (*p* = 0.005) with 79% sensitivity and 70% specificity of miR-483-5p *p* ≤ 0.77 for detection of CAD (Figure 2).

### 2.4. Cardiovascular Disease-Related Medications and Their Association within the Study Groups

We performed statistical analysis for the patients’ treatment regimen who came to hospital with the following treatment medications: beta blockers, angiotensin receptor blockers (ARBs), calcium channel blockers (CCBs), angiotensin-converting enzyme (ACE) inhibitors, anticoagulants, aspirin and statins (see Table 3). Overall, the number of patients who were on various medications varied, with most patients being on aspirin and statins. There was a trend towards medication intake in the control group with patent arteries, as well as in the chronic CAD group, who underwent percutaneous coronary intervention. After ANOVA, differences among the groups were seen with beta blockers, aspirin and angiotensin converting enzyme (ACE) inhibitors (*p* = 0.006, *p* < 0.001, *p* = 0.044, respectively). After pairwise analyses, there were significantly more CAD patients on beta blockers (lowering blood pressure) compared to controls (57% vs. 26%), and much fewer STEMI patients on aspirin (antithrombotic) compared to controls (18% vs. 65%). There were much fewer STEMI patients on beta blockers and aspirin compared to CAD patients (9% vs. 57% and 18% vs. 79%, respectively), and much fewer NSTEMI patients on ACE inhibitors and aspirin compared to CAD patients (8% vs. 50% and 26% vs. 79%, respectively). There were significant differences with aspirin between all groups.

Further, we performed statistical analysis to identify whether there is a difference in miR-483-5p expression in serum samples of patients treated with either beta blockers, ACE inhibitors or aspirin versus patients who were not on these treatments. Interestingly, we noticed that patients receiving aspirin had significantly higher expression levels of miR-483-5p, both in peripheral and in culprit coronary serum samples (*p* = 0.0168 and *p* = 0.0244, respectively) (Figure 3).

### 2.5. Predicted Gene Target for miR-483-5p

Table 4 lists all the predicted gene targets of miR-483-5p in cardiac tissue with the indication of Reads Per Kilobase of transcript, per million mapped reads (RPKM). Out of 59 predicted targets summarized in the miRDB website, 14 expressed in the heart. The listed genes are involved in inflammation (*PLA2G5*, biased expression in heart, RPKM 14.5), mitochondrial regulation and oxidative stress (*NUDT8*, *GRK2*), sarcomere (*MYOM2*, biased expression in heart, RPKM 193.6), cytoskeletal (*IQSEC2*) and cell morphogenesis (*ZMYM6*). Some gene targets were zinc finger proteins involved in transcriptional regulation (*ZNF584*, biased expression in many tissues—amongst them higher expression in the heart, RPKM 1.4, and *ZNF417*) and RNA replication (*SMG6*). Other genes were involved in apoptosis (*DNAAF10*), inhibition of extracellular matrix degradation and endothelial proliferation (*TIMP2*) and connective tissue regulation (*ADAMTS2*), as well as genes promoting angiogenesis (*HGSNAT*).

## 3. Discussion

In our current study, we found that compared to chronic CAD patients, acute MI patients had a significant upregulation of miR-483-5p demonstrated by a high ‘coronary arterial gradient’ at the site of the acute occluded culprit artery. This was in contrast to the significantly lower levels of miR-483-5p in the systemic peripheral arterial circulation in both groups of patients compared to controls without significant CAD.

CAD mortality and prevalence vary among countries, and estimation of the true prevalence of CAD in the population is complex [17]. Although developing countries show considerable variability in the incidence of CAD [17], our study showed no differences between Israeli ethnic groups of Jewish and Arab populations in all groups studied. This may largely be due to similarities in genetic background, geographical habitats and healthcare systems.

We saw significant differences between the groups with beta blockers, ACE inhibitors and aspirin. These medications lower blood pressure while aspirin is a known antithrombotic. Interestingly, aspirin was the only drug that associated positively with miR-483-5p expression in peripheral blood and in blood at the occluded artery in all patients (Figure 3). Aspirin is an analgesic and anti-inflammatory drug, and acts through other mechanisms in the prevention of cardiovascular disease, such as platelet inactivation [18]. We can suggest that miR-483-5p may play an important role in platelet inactivation.

Recently, Belland Olsen et al. [19] showed the targeting of new medications on inflammatory mediators, known as the inflammasomes, is effective in treating CAD patients. These medications are Colchicine, Canakinumab, and Anakinra. Another study in cultured human pulmonary arterial endothelial cells showed that overexpression of miR-483-5p inhibited inflammatory and fibrogenic responses, revealed by the decreased expression of TGF-β, TGF-β -receptor-2, β-catenin, connective tissue growth factor, interleukin-1b and endothelin-1 [20]. The above results may suggest a critical role for the regulation of these genes with miR-483-5p in vivo, inhibiting the pathogenic mechanisms in heart disease.

Our results may suggest that miR-483-5p may be released from endothelial cells of blood vasculature at the occluded area as part of a protection mechanism after an acute MI event (Figure 1). Indeed, overexpression of miR-483-5p in cultured human pulmonary arterial endothelial cells inhibits inflammatory and fibrogenic responses, as revealed by the decreased expression of transforming growth factor-β, transforming growth factor-β receptor 2, β-catenin, connective tissue growth factor, interleukin-1b, and endothelin-1 [20].

We can speculate and suggest that adjacent tissues and cells absorb the locally abundant miR-483-5p, thus making it a paracrine modulator (Figure 2). This is supported by the finding that circulating miR-483-5p is found in exosomes, extracellular vesicles playing an essential role in intercellular communication [3,21,22,23]. Activated endothelial cells at the culprit area make good candidates for exosomes, including miR-483-5p-containing exosome recipients. Local increases in miRNA-483-5p might be a part of a protective mechanism intended to limit the production of pro-inflammatory mediators, oxidative stress agents and calcification factors. Yet, excessive levels of this miRNA molecule at the culprit area can potentially have a harmful effect and promote hypoxia-induced apoptosis, as well as inhibit angiogenesis and wound healing [24,25]. MiR-483-5p overexpression in human fibroblasts and endothelial cells promoted the expression of fibrosis-related genes [21]. Thus, we can assume, based on previous studies and ours, that the miR-483-5p molecule could have protective as well as deleterious effects on-site of the injury in MI patients, depending on the concentration gradient.

In the present study, we specify the candidate target genes for mir-483-5p in heart tissue, among them *PLA2G5*, *GRK2*, *MYOM2*, *TIMP2*, *ADAMTS2* and *HGSNAT.* Indeed, regulation of many of these genes related to cardiovascular diseases and correlated with MI in previously published reports [26,27,28,29,30,31] (Figure 2). For instance, elevated plasma sPLA2-IIA (encoded by the *PLA2G5* gene) predicts coronary heart disease risk [26] and is involved in inflammation. *GRK2* was suggested as a therapeutic target for heart failure, as its genetic inhibition initiated heart protection of adverse remodeling [27]. The *MYOM2* gene, which encodes Myomesin-2 protein, is a major component of the sarcomere that maintains contractility in heart muscle; additionally, it is involved in heart development [28]. Therefore, *MYOM2* may be important for hypertrophic cardiomyopathy and Tetralogy of Fallot [28]. *MYOM2* is downregulated in both animal models and patients with phenotypes related to major adverse cardiovascular events [29]. The inhibitory function of another miR-483-5p gene target revealed by our analysis is *TIMP2*, a key determinant of post-MI myocardial remodeling; *TIMP2* replenishment in diseased myocardium could provide a potential therapy in reducing or preventing disease progression [30]. *ADAMTS2* may serve as a promising target for preventing cardiac hypertrophy and heart failure [31]. Lastly, the *HGSNAT* gene, discovered in our in silico analysis as another gene target of miR-483-5p, functions in lysosomal degradation of heparin sulfate, and heparin sulfate degradation is known to promote tissue repair and angiogenesis after acute MI [32]. We speculate from the above-mentioned that the *HGSNAT* gene promotes tissue repair and angiogenesis locally in the culprit area after acute MI. Our results also propose that *PLA2G5* reduces inflammation, and that *GRK2 MYOM2 TIMP2* and *ADAMTS2* regain and maintain normal heart function, cell morphogenesis and metabolism following the injury of acute MI (Figure 2). These hypotheses need further validation.

Several studies address miR-483-5p expression in hypercholesterolemia and CAD patients. MiR-483-5p was demonstrated to be under-expressed in hypercholesterolemia patients, whereas statin treatment can reverse its expression to normal levels [33], thus supporting our findings. In MI, miR-483-5p was previously shown as upregulated compared with a control group [34]. Different demographic populations recruited to these studies can explain this disagreement with our study. In addition, whereas in previous reports results yielded only from peripheral venous blood, we collected blood samples both from peripheral arterial circulation and from coronary arteries simultaneously. In our current study, all control group participants had no history of either coronary or known myocardial disease and, in addition, all underwent diagnostic coronary angiography to exclude CAD. This is different from other studies in which the control group was composed of patients either with non-cardiac chest pain [34] or healthy volunteers after a standard medical examination, so we believe our control group was a more homogeneous one.

Although both positive and negative effects of miR-483-5p were demonstrated in cardiac disease, it is commonly believed that under normal conditions miR-483-5p has a protective role in preventing cardiac diseases including CAD, as it moderates inflammation, oxidative stress, pro-inflammatory cytokine-induced apoptosis and fibrosis, as well as regulated endothelial function [20,33,35,36,37].

Our results imply that miR-483-5p under pathologic conditions is relatively low, while ‘boosts up’ as a protective mechanism at the culprit occluded artery following acute MI. High miR-483-5p expression may be induced by the release of miR-483-5p from endothelial cells at the site of local ischemia (Figure 1 and Figure 2). The schemes further show that intracellular miR-483-5p may inhibit deleterious genes and may allow activation of protection genes.

Due to its many gene targets, miR-483-5p may serve as a good candidate biomarker for further understanding and studying cardiovascular disease. Further molecular studies are required to characterize miR-483-5p cellular functions and valid target genes in these pathophysiological processes. The resultant gene targets discussed in this study may improve our knowledge of the mechanisms involved and may guide novel therapeutic approaches. We suggest that protective mechanisms induced by miR-483-5p are disrupted in CAD, as levels of miR-483-5p are significantly lower in CAD patients (Figure 1). Figure 2 demonstrates a protection mechanism at the culprit vessel in acute MI patients.

There are a few potential miRNA-targeted emerging therapies for cardiovascular disease. Recently, it has been shown that therapeutic delivery of microRNA-125a-5p oligonucleotides improves recovery from myocardial ischemia/reperfusion injury in mice and swine [38]. Moreover, miR-423-5p inhibition exerts protective effects on angiotensin II-induced cardiomyocyte hypertrophy [39]. Furthermore, it was recently shown that endothelial miR-483-3p, the other strand of miR-483-5p, is hypertension protective [40]. These novel findings are exciting and certainly encourage us to continue our validations for targeted miR-483-5p therapies in heart disease.

We will address a few limitations to our study. The cohort groups were relatively small and focused on male subjects only. Because of Research Ethics Board limitations, we did not sample blood from the non-culprit coronary artery, nor did we repeat coronary sample from the previous occluded artery.

Our study focused on male patients solely for the following two main reasons. First, we anticipated general differences in circulating miRNAs between males and females, where in females the expectation is for “noisier” outcomes due to hormonal factors. Lamon et al. very recently showed that gene targets of circulating miRNAs varied considerably with the menstrual cycle and were primarily involved in cell proliferation and apoptosis [41]. This exploratory study suggests that circulating miRNAs may play an active role in the regulation of the female cycle by mediating the expression of genes during fluctuating hormonal changes. In this case, finding differences between the groups in our study will be much more difficult. Second, during the patient recruitment process we noticed that female patients were almost absent in the STEMI group. In this case, including females in all study groups making them homogeneous would be practically impossible.

The rationale for collecting samples prior to the intervention was to identify miRNAs specifically enriched in the microenvironment of the blocked artery. We assumed that after intervention this microenvironment does not exist any longer.

Unfortunately, we do not have information about clinical characteristics such as CRP and HbA1C. Troponin was checked in all NSTEMI and some STEMI patients, no Troponin tests were performed in control patients and CAD patients since there is no medical indication for that. It is expected that myocardial damage is higher in NSTEMI and STEMI patients compared to control and CAD patients. We agree that we cannot exclude the work by Li et al. who showed high association between circulating miR-497 and acute myocardial infarction, which was also correlated with high troponin levels [42].

## 4. Materials and Methods

### 4.1. Patient Recruitment and Sample Collection

All procedures in the study involving human subjects were performed in compliance with the institutional guidelines of Poriya Medical Center, Israel, and were approved by the Helsinki Committee (ethical approval No. POR-0017-14). Serum samples were obtained from male patients undergoing cardiac catheterization procedures in the cardiovascular department of Poriya Medical Center, Israel, after signing informed consent forms. Four groups of patients were recruited to the study: 22 acute STEMI, 26 acute NSTEMI, 28 chronic CAD patients and 23 control patients who had no previous history of coronary or myocardial disease, no previous coronary intervention, and current coronary angiogram-demonstrated patent coronary arteries. In most cases, coronary angiography was performed on patients with patent coronary arteries to exclude CAD due to patients’ complaints/ECG changes combined with relevant risk factors. Aortic stenosis patients were not excluded in this study. Patients were defined as patent if no occlusion, or occlusion of less than 50%, was observed in the coronary tree. Exclusion criteria were as follows: female sex, younger than 18 years of age, patients who underwent coronary artery bypass surgery, patients with an active oncological disease, patients on any form of dialysis or under antibiotics, steroids or other anti-inflammatory modulators treatment or having evidence of systemic inflammation or infection. Specifically, in the control group, previous myocardial disease and/or previous coronary intervention were excluded on top of their current coronary angiogram, which had to demonstrate patent coronary arteries.

Two blood samples were obtained from each patient during the coronary catheterization procedure, one from each of the peripheral and the coronary arterial system. Specifically, in acute STEMI, NSTEMI and chronic CAD patients, coronary blood samples were taken just prior to the coronary intervention from the hemodynamically significant culprit coronary artery narrowing (>70%), related to the patient’s manifestation of symptoms. In the control patients with the patent coronary arteries, coronary samples were collected from a random artery. Blood samples were collected into serum-separating tubes, kept at room temperature and were processed within 24 h. To obtain serum, samples were centrifuged for 15 min at 2000× *g*. The obtained serum samples were aliquoted into Eppendorf tubes and stored at −80 °C until RNA extraction.

### 4.2. RNA Extraction

RNA was extracted from 1 mL of serum using the miRNeasy Serum/Plasma kit (Qiagen, Hilden, Germany). RNA concentration was measured with Qubit miRNA assay kit (Molecular Probes Life Technology, Thermo Fisher Scientific Inc., Waltham, MA, USA). The quality of extracted RNA was assessed by Bioanalyzer and TapeStation sytems for small RNA quantitation, purity and integrity.

### 4.3. Preparation of Libraries and miRNA Sequencing

MiRNA libraries were prepared from RNA samples of 16 patent and 18 STEMI patients, using the QIAseq miRNA Library Kit (Qiagen, Hilden, Germany). Library concentrations were measured with Qubit dsDNA HS Assay Kit (Molecular Probes, Life Technology, Thermo Fisher Scientific, Waltham, MA, USA) and the quality was assessed by Bioanalyzer and TapeStation systems (Agilent Technologies, Santa Clara, CA, USA). All miRNA libraries were sequenced using Nextseq 550 (Illumina, San Diego, CA, USA) with a read length of 75bp.

### 4.4. Quantitative Real Time PCR (qPCR)

Complimentary DNA was synthesized using the TaqMan Advanced miRNA cDNA Synthesis Kit (Applied Biosystems, Life Technologies, Thermo Fisher Scientific, Waltham, MA, USA). For the qPCR reaction, TaqMan Advanced miRNA RT-PCR kit was used (Applied Biosystems, Life Technologies, Thermo Fisher Scientific, Waltham, MA, USA). For each reaction, 5 µL aliquots of diluted cDNA were used, and miR-1260a-5p was used as a reference [43]. Specific miRNA probes were obtained from Applied Biosystems (Thermo Fisher Scientific, Waltham, MA, USA): has-miR-1260a (Assay ID: 478476_miR) ahashsa-miR-483-5p (Assay ID: 478432_miR). The qPCR reactions were performed using CFX-Connect Real-Time System (Bio-Rad, Hercules, CA, USA) and the following conditions: 95 °C for 20 s, 95 °C for 3 s and 40 cycles, 60 °C for 30 s, 65 °C for 5 s and a final cycle at 95 °C for 50 s. The ΔΔCt method was used for the real-time PCR quantification analysis.

### 4.5. Bioinformatics and Statistical Analysis

Primary and secondary analysis of miRNA sequencing results was performed using a web resource Gene Globe Data Analysis Center (Qiagen, Hilden, Germany). The Trimmed Mean of M (TMM) value normalization method was used for the differential expression analysis.

The *t*-test for independent samples was performed to compare clinical characteristics of the patients in each of the four study groups; *p*-values are shown in a table. In the qPCR experiment, the Kruskal–Wallis test for independent samples was performed, followed by post hoc Dunn’s and Conover’s analysis. For testing the ability of miR-483-5p to differentiate between patent and chronic CAD, receiver operating characteristic (ROC) curve analysis was performed. All the statistical tests were performed using MedCalc^®^ software (MedCalc Software Ltd, Ostend, Belgium), version 19.5.3.

### 4.6. Identification of Predicted Gene Targets for miR-483-5p

miRDB is an online database for miRNA target prediction and functional annotations [44,45]. All the targets in miRDB were predicted by a bioinformatics tool, MirTarget, which was developed by analyzing thousands of miRNA–target interactions from high-throughput sequencing experiments. Common features associated with miRNA binding and target downregulation were identified and were used to predict miRNA targets with machine learning methods. These miRNAs, as well as associated functional annotations, are presented in the FuncMir Collection in miRDB. As a recent update, miRDB presents the expression profiles of hundreds of cell lines and the user may limit their search for miRNA targets that are expressed in a cell line of interest. To facilitate the prediction of miRNA functions, miRDB presents a new web interface for integrative analysis of target prediction and gene ontology data.

Using the miRDB website, predicted gene targets for miR-483-5p are available for studying different tissues, including the heart. We distinguished gene targets and mechanisms applicable to chronic and acute MI states using this site and the NCBI websites, Gene Ensembl.

## 5. Conclusions

To the best of our knowledge, for the first time we show herein significant association between miR-483-5p downregulation and heart disease, in peripheral serum of acute MI and chronic CAD patients. Yet, our acute MI patients had significantly higher miR-483-5p levels locally at the culprit occluded coronary artery compared with chronic CAD patients. Our data suggest that miR483-5p is pathologically under-expressed, and this local regulation at the site of the culprit artery suggests a potential attempt for on-site protection. These results may offer novel leads for the mechanism in which specific miRNAs are involved in the pathogenesis and healing process in both chronic and acute manifestations of CAD.

## Data Availability

NGS data were deposited to public database GEO and is available at https://www.ncbi.nlm.nih.gov/geo/query/acc.cgi?acc=GSE230165, accessed on 23 April 2023.

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
