# Peer review of "Characterization of Systemic and Culprit-Coronary Artery miR-483-5p Expression in Chronic CAD and Acute Myocardial Infarction Male Patients"

_ijms, 2023, doi:10.3390/ijms24108551_

Round 1

Reviewer 1 Report

Great introduction section.

Please consider discussing the patients with CAD, STEMI and Non-STEMI who are often on life-long HF, hypertension, and high-cholesterol therapies such as -statin, ACE Inhibitors, beta-blockers, CCBs and more. These are likely to regulate the cardiac remodelling and expression of miR-483-5p, other miRNAs and regulated genes. The HF therapies with vaso-dilation or constriction therapeutic pathways, directly affect the expression of genes such as TGF-β, IL-1b, and endothelin-1 and therefore the associated miRNAs.

Further discussion on the demographics of the patients could be beneficial since the CAD has significant data available based on ethnicity and demographics. 

Overall this is a great study. Further potentials of mi-RNA-targeted emerging therapies for HF can be discussed with emphasis on miR483-5p being a potential target of further studies. 

Reviewer 2 Report

In the manuscript entitled “Characterization of systemic and culprit-coronary artery miR-483-5p expression in chronic CAD and acute myocardial infarction patients” authors compared microRNA expression in chronic-CAD and acute-MI patients in peripheral blood vasculature versus coronary arteries proximal to a culprit area. In the study they showed high miR-483-5p ‘coronary arterial gradient’ between acute-MI chronic-CAD. According to abovementioned results they hypothesize that miR483-5p play important role in mechanisms related with local myocardial ischemia.

The paper is well-written, however study design requires some explanations. Discussion is based on large literature review. The role of miR-483-5p was described in myocardial infarction, however the study has some novelty not in blood samples from coronary arteries. However there some issues that need to be addressed.  

1.      What was the cause of coronary angiography in patients with patent coronary angiography? Did you exclude cases with aortic stenosis?

2.      Why the study was provided only in male. Are you expecting that protective mechanisms related to miR483-5p may differ in men and women? I would rather recommend to change in the title “myocardial infarction patients” to “ men with myocardial infarction”?

3.      Why blood samples from coronary arteries were taken just once and prior the intervention not after during reperfusion? In my opinion the second situation – reperfusion would be mu more valuable.

4.      There is no information about the treatment prior hospitalization. This should be discussed in the context of relation between treatment and MiR483-5p.

5.      Authors should provide detailed information about clinical characteristic of patients with MI and control groups at admission (CRP, HbA1C, EF, CTNI,  presence of heart failure, cardiogenic shock, blood pressure on admission) as in this case we can not conclude that it is related only to MI or maybe to heart failure? Li et al. (2014) found correlation between miR-489 (from peripheral blood samples) and troponins levels. And troponins levels are usually related to myocardial damage. This should be also discussed.  

Li, Z., Lu, J., Luo, Y., Li, S., and Chen, M. (2014). High association between human circulating microRNA-497 and acute myocardial infarction. Sci. World J. 2014:931845. doi: 10.1155/2014/931845

6.      Line 179 ‘heart attack” is rather colloquialism.  “infarct-related artery” would be more suitable.

7.      Did you plan to follow-up the study population, as it would be interesting to compare miR483-5p with myocardial damage in patients with STEMI, in particular.
